# Sarcopenia is Independently Associated with an Increased Risk of Peptic Ulcer Disease: A Nationwide Population-Based Study

**DOI:** 10.3390/medicina56030121

**Published:** 2020-03-11

**Authors:** Youn I Choi, Jun-Won Chung, Dong Kyun Park, Kwang Pil Ko, Kyung Oh Kim, Kwang An Kwon, Jung Ho Kim, Yoon Jae Kim

**Affiliations:** 1Department of Gastroenterology, Gil Medical Center, Gachon University, Incheon 21565, Korea; cys7like@hanmail.net (Y.I.C.); pdk66@gilhospital.com (D.K.P.); kkoimge@gilhospital.com (K.O.K.); toptom@gilhospital.com (K.A.K.);; 2Department of Preventive Medicine, Gachon University College of Medicine, Incheon 21565, Korea; kpko@gachon.ac.kr

**Keywords:** gastric ulcer, duodenal ulcer, obesity, sarcopenia

## Abstract

*Background and objective:* Although obesity is associated with an increased risk of peptic ulcer disease (PUD), no study has evaluated the association of PUD with sarcopenia. The aim of this study was to evaluate the association of sarcopenia and obesity with PUD. *Material and Methods:* Data from the Korean National Health and Nutrition Examination Survey (KNHANES) IV and V for 2007–2012 were used. PUD history, dietary, alcohol consumption, smoking, physical activity patterns, and other socioeconomic factors were analyzed. Sarcopenia index (appendicular skeletal muscle mass (kg) ÷ body mass index (kg/m2)) and body fat mass were determined by dual-energy X-ray absorptiometry. Univariate and multivariate analyses were performed to evaluate the association of sarcopenia with the prevalence of PUD. *Results:* The 7092 patients were divided into the sarcopenic obesity (SO, n = 870), sarcopenic non-obesity (n = 2676), non-sarcopenic obesity (NSO, n = 2698), and non-sarcopenic non-obesity (NSNO, n = 848) groups. The prevalence of PUD in these groups was 70 (7.9%), 170 (7.4%), 169 (6.3%), and 47 (3.8%), respectively (*p* < 0.001). A crude analysis revealed that the prevalence of PUD was 2.2-fold higher in the SO group than in the NSNO group (odds ratio (OR), 2.2; 95% confidence interval (CI), 1.5–3.2), the significance of which remained after adjustment for age, sex, body mass index, and HOMA-IR (homeostatic model assessment insulin resistance) score (OR, 1.9; 95% CI, 1.3–2.7). *Conclusion:* In conclusion, in this nationally representative cohort, the combination of muscle and fat mass, as well as obesity, was associated with an increased risk of PUD.

## 1. Introduction

Peptic ulcer disease (PUD) imposes a substantial burden in terms of its symptoms, impairment of the health-related quality of life, and costs [1,2]. Determination of the etiology and pathogenesis of PUD would facilitate its prevention and management [1,2].

The complex and multifactorial pathogenesis of PUD is related to various risk factors [1,2,3]. An imbalance of gastric luminal factors and disruption of the barrier function of the mucosa are involved in the pathogenesis of PUD [1,3]. Diverse environmental factors—including *Helicobacter pylori* infection, smoking, excessive alcohol consumption, and a variety of drugs—are ulcerogenic [1,3,4,5]. Recent reports have also linked metabolic inflammation, such as obesity, with susceptibility to mucosal injury, including PUD [1,3].

Sarcopenia is also closely related to metabolic inflammation [6,7,8,9]. Sarcopenia is characterized by progressive and generalized loss of the mass and function of skeletal muscle [10,11,12,13]. Sarcopenia is categorized into primary and secondary sarcopenia. Primary sarcopenia is a disease of the elderly, malnourished, and neglected [7,13]. In contrast, secondary sarcopenia occurs in relatively young people and is associated with chronic inflammatory conditions [11,13]. Sarcopenia has been related with diverse disease outcomes including inflammatory bowel disease [14], cardiovascular disease [9,14,15], liver disease [16,17,18], lung and kidney disease [19], frailty [8,20], cognitive dysfunction [21], depression [21], and almost all types of malignancies [22,23,24].

Although metabolic health is associated with the development of PUD [3,6], and sarcopenia is associated with the prognosis of gastrointestinal diseases, whether sarcopenic status is associated with the development of PUD has not been assessed.

Herein, we explored whether PUD is associated with sarcopenia using data from the Korean National Health and Nutrition Examination Survey (KNHANES).

## 2. Methods

### 2.1. The Korea National Health and Nutrition Examination Survey

Data were collected from the KNHANES IV (2007–2009) and V (2010–2012) annual nationwide cross-sectional surveys of the health and nutrition status of the Korean population [25]. The KNHANES IV and V were conducted by the Korea Center for Disease Control and Prevention (KCDC) [26]. All survey protocols were approved by the Institutional Review Board of the KCDC [26], and written informed consent was obtained from all subjects. This study was conducted in accordance with the Declaration of Helsinki [26]. The original KNHANES data are de-identified and publicly available on the KNHANES website (http://knhanes.cdc.go.kr) [26]. The KNHANES IV data contain detailed information on (1) socioeconomic characteristics including income, educational level, employment, and experience; (2) health behaviors including alcohol consumption, smoking history, physical activity, and sleep duration; (3) medical history including diabetes, hypertension, metabolic syndrome, cardiovascular disease, neuromuscular disease, PUD, and malignancy; (4) medication history including for cardiovascular and neuromuscular disorders; (5) anthropometric indices including weight, height, waist circumference, and BMI; and (6) laboratory parameters including the levels of fasting blood glucose, hemoglobin A1C (HbA1C), serum insulin, high-density lipoprotein (HDL), and low-density lipoprotein (LDL) [25]. Further details of the KNHANES VI can be found on the website (http://www.knhanes.cdc.go.kr).

### 2.2. Study Population

We used the KNHANES data from 2007 to 2012. Among 50,405 potential subjects, those with any of the following were excluded: age < 19 years (n = 12,400), missing information on PUD history (n = 30,099), and missing data on fat or muscle mass (n = 814). Finally, 7092 subjects were included in this study.

### 2.3. Measurement of Muscle Mass and Fat Mass

Height, weight, and waist circumference were determined in accordance with the standard international guidelines [25]. Body mass index (BMI) was calculated as body weight (kg) divided by height (m) squared (i.e., kg/m^2^). The homeostatic model assessment insulin resistance (HOMA-IR) [27] score of insulin resistance was calculated as: *HOMA-IR =* fasting blood glucose (mmol = L) × insulin (µU = mL) ÷ 22.5.

Body composition, including body fat and appendicular skeletal muscle mass (ASM), was assessed by dual-energy X-ray absorptiometry (DISCOVERY-W Fan-Beam Densitometer; Hologic Inc., Bedford, MA, USA) [25]. ASM was defined as the sum of the muscle mass in the arms and legs, assuming that all non-fat and non-bone tissues were skeletal muscles [13]. We defined sarcopenia as one of the following: (1) percentage muscle mass (ASM ÷ weight^2^) below the lowest quintile of the study population or within one standard deviation of that of a healthy young population (‘Appendiceal muscle mass weight’); (2) percentage muscle mass (ASM ÷ height^2^) below the lowest quintile of the study population or within one standard deviation of that of a healthy young population (‘Appendiceal muscle mass height’); or (3) percentage muscle mass (ASM ÷ BMI) below the lowest quintile of the study population or within one standard deviation of that of a healthy young population (‘Appendiceal muscle mass BMI’). These are based on the recommendations of the consensus report of the Asian Working Group for Sarcopenia [13].

Fat mass was defined as the sum of the fat mass of all body parts. The high-fat mass group was defined as the patients in the highest quintile of fat mass.

To assess the prevalence of PUD, we divided the subjects into low-muscle high-fat (sarcopenic obesity; SO), high-muscle high-fat (non-sarcopenic obesity; NSO), low-muscle low-fat (sarcopenic non-obesity; SNO), and high-muscle low-fat (non-sarcopenic non-obesity; NSNO) groups.

### 2.4. History of PUD

The history of PUD was assessed based on self-reported data [25]. The subjects were categorized as having PUD if they responded positively to the question, “Do you have clinician-diagnosed peptic ulcer disease including gastric or duodenal ulcer?” [25].

### 2.5. Physical Activity, Smoking, and Alcohol Consumption

Physically active subjects were defined as those who engaged in physical activity for at least 150 min per week, medium- or high-intensity physical activity for 75 min per week, or in both medium- and high-intensity physical activity for 60 min per week [25].

Smoking status was assessed by a self-administered questionnaire and was categorized as never-smoker, ex-smoker, or current smoker [28]. Alcohol consumption was assessed by a self-administered questionnaire [28]. Participants were asked about lifetime and current alcohol consumption status and the volume of alcohol consumed on any one occasion [25,28]. Subjects who consumed alcohol more than once per month were defined as current drinkers [25,28]. Subjects who consumed seven or more drinks on one occasion for men, or five or more drinks for women were defined as heavy drinkers [25,28].

### 2.6. Statistical Analysis

Statistical analyses were conducted using SPSS for Windows version 20.0 (IBM Corporation, Armonk, NY, USA). Differences in demographic and anthropometric characteristics according to sarcopenia and PUD were compared using Student’s *t*-test or χ^2^ test. To assess the relationship between sarcopenic obesity and PUD, fasting glucose, HbA1c, insulin levels, HOMA-IR score, dietary pattern, physical activity, smoking, alcohol consumption, and socioeconomic status were determined, and a logistic regression analysis was performed after adjusting for several variables. Models were initially run after adjusting for age, sex, and BMI (model 1) and were repeated after adding other variables (models 2 to 6).

A value of *p* < 0.05 was taken to indicate statistical significance. Continuous and categorical variables are expressed as means ± standard deviation and n (%), respectively.

## 3. Results

### 3.1. Clinical Characteristics according to Sarcopenic Status

Age, educational level, household income, and marital status differed significantly between the sarcopenic and non-sarcopenic groups (all *p* < 0.05) (Table 1). Although health behaviors are associated with sarcopenic status, there was no significant difference in current smoking status (*p* = 0.2) or regular exercise frequency (*p* = 0.2) between the two groups. The non-sarcopenic group showed a higher prevalence of alcohol consumption (*p* < 0.05) (Table 1). The sarcopenic group had a higher prevalence of metabolic syndrome and cardiovascular disorders (*p* < 0.05). The sarcopenic group had a higher body fat mass, BMI, and waist circumference than the non-sarcopenic group. Interestingly, the sarcopenic group had a higher prevalence of a history of PUD than the non-sarcopenic group (n = 240, 7.6%; *p* = 0.001).

### 3.2. Obesity and Sarcopenic Indices according to the Presence of Peptic Ulcer Disease according to History of PUD in Univariate Analysis

The prevalence of a history of PUD differed significantly between the SO vs. NSO groups and the NSO vs. SNO groups (*p* < 0.05; Table 2; Table 3, and Figure 1). In univariate analysis, patients with PUD had more prevalence of SO and SNO than those with no previous history of PUD group (SO; 15.4% vs. 12.3%, SNO; 37.3% vs. 32.0%) (Table 3).

A positive linear trend in the prevalence of PUD was found in the SO, SNO, NSO, and NSNO groups (*p* < 0.001; Figure 1).

### 3.3. Odds Ratios for Risk Factors for PUD in Multivariate Anlaysis

After adjustment for age, sex, BMI, and HOMA-IR (model 2), the prevalence of PUD in the SNO group was 1.4-fold higher than that in the NSNO group, and the PUD prevalence in the SO group was 1.9-fold higher than that in the NSNO group; these differences were significant (Table 4).

## 4. Discussion

In this nationally representative population-based study, we demonstrated that among patients with obesity, those who also have sarcopenia have a higher prevalence of PUD. The subjects in the SO group had a higher incidence of PUD and showed a linear-by-linear association of PUD prevalence across groups SO, SNO, NSO, and NSNO (*p* < 0.001). The combination of an increased fat mass and decreased muscle mass was associated with the risk of PUD. These associations remained evident even after adjusting for established risk factors, such as sex, old age, alcohol consumption, smoking, medications for cardiovascular and musculoskeletal disease, and perceived stress.

The prevalence of PUD was significantly higher in the SO than in the SNO group (*p* < 0.05). Therefore, not only the obesity index, but also the sarcopenia status is associated with the prevalence of PUD.

The mechanisms underlying the association of sarcopenia and PUD are unclear, but there are several candidates. First, sarcopenic obesity is related to increased levels of circulating oxidative stress markers, which lead to alterations in the biological properties of the membrane and to amplification of cellular damage [9]. Bellanti reported that circulating levels of oxidative stress markers are particularly increased in SO compared to NSO patients [9]. Sarcopenic patients might have inadequate reserves to recover from mucosal injury. Second, sarcopenia may be associated with an altered gut microbiome [29]. The skeletal muscle–gut axis is emerging mechanisms of sarcopenia for intestinal disease [29]. The pathogenesis of sarcopenia involves systemic inflammation, impaired muscle protein synthesis, increased muscle cell apoptosis, mitochondrial dysfunction in skeletal muscle tissue, and insulin resistance [29]. The gastrointestinal tract and the gut microbiota are associated with muscle function and metabolism, although the mechanisms underlying this link are unclear [29].

To our knowledge, this is the first population-based study to assess the association of sarcopenia with the prevalence of PUD. Prior studies of the association of PUD with obesity indices, including BMI, waste circumference (WC), and fat mass, were reportedly controversial [3,30,31]. A national Taiwanese survey of the associations of BMI with gall stones, fatty liver disease, chronic viral hepatitis, hemorrhoids, gastrointestinal polyps, and PUD showed that BMI was not associated with the prevalence of PUD in men or women [30]. A prospective cohort study in the US demonstrated that obesity indices are related to an increased prevalence of gastric, but not duodenal, ulcers [7]. A national survey in Korea regarding the association of the obesity index with PUD reported that in women, weight, hip circumference, and BMI are associated with PUD, and hip circumference in men [3]. The controversy regarding the effect of the obesity index on PUD might be because prior studies did not consider the effect of muscle mass. In this study, we assessed not only fat mass but also muscle mass, and the combination of the fat mass and sarcopenia indices was significantly associated with the prevalence of PUD.

According to recent reviews, idiopathic ulcers comprise 10–30% of PUD cases, and this proportion has recently increased in Asia [32,33]. The increasing prevalence of sarcopenia indicates it to be a candidate cause of idiopathic ulcers.

Sarcopenia has diverse disease-related outcomes, for instance, postoperative complications in gastric cancer, rectal cancer, and pancreatic cancer; hepatocellular carcinoma; post-bariatric surgery in severely obese patients; outcomes of chemotherapy for pancreatic cancer [23]; and an increased mortality rate for patients with chronic kidney disease [16,34]. However, the association between PUD and sarcopenia has not been investigated.

Medical therapies for PUD target gastric acid secretion and mucosal defense mechanisms. Obesity and metabolic syndrome are linked to mucosal defense mechanisms, which are related to the development of PUD. In this study, sarcopenia and visceral fat mass index were related to the prevalence of PUD. We found a non-linear relationship between muscle mass and fat mass, suggesting that these have independent impacts on the prevalence of PUD. Our finding that sarcopenia is independently associated with PUD suggests that the latter can be prevented by increasing muscle mass.

This study had several limitations. First, PUD was defined as a previous diagnosis by a physician. Endoscopic findings were not available. However, the incidence of endoscopically confirmed PUD is 2–10% in Korea, and in this study, the prevalence of PUD was 3.8–7.9%, in line with previous reports [33]. Therefore, we probably did not overestimate the prevalence of PUD [33]. Moreover, all Koreans more than 40 years of age undergo a health examination for cancer screening; this includes an upper endoscopy based on the recommendation of the Korean surveillance guideline for endoscopy at 1- or 2-year intervals. Second, we quantified only the skeletal muscle mass; its function was not evaluated. According to a recent report on the revised European consensus regarding the definition and diagnosis of sarcopenia, definite diagnosis of sarcopenia includes the presence of low muscle quantity or quality [35]. Severe sarcopenia is considered when low muscle strength, low muscle quantity/quality, and low physical performance are detected [35]. Since we did not measure muscle quality or muscle strength among the study population, the interpretation of our results should be regarded with caution. Further large population-based studies measuring both muscle quality and quantity should be done. Third, PUD risk factors such as *Helicobacter pylori* infection or treatment history were not available. Fourth, drug histories (including of antiplatelet agents or non-steroidal anti-inflammatories) were not available. Instead, we evaluated the patients’ self-reported history of medications for cardiovascular and musculoskeletal disease. Despite these limitations, the strength of this study lies in the assessment of the effect of muscle mass and fat mass on the prevalence of PUD in a nationally representative sample.

In conclusion, obesity status and the combination of muscle and fat mass are associated with an increased risk of PUD. Sarcopenic obesity was associated with a higher prevalence of PUD, suggesting the need for endoscopic verification in patients with a low-muscle mass and high-fat mass, and that sarcopenia is a candidate cause of idiopathic PUD. Further large-scale multicenter studies should evaluate this hypothesis.

## Figures and Tables

**Figure 1 medicina-56-00121-f001:**
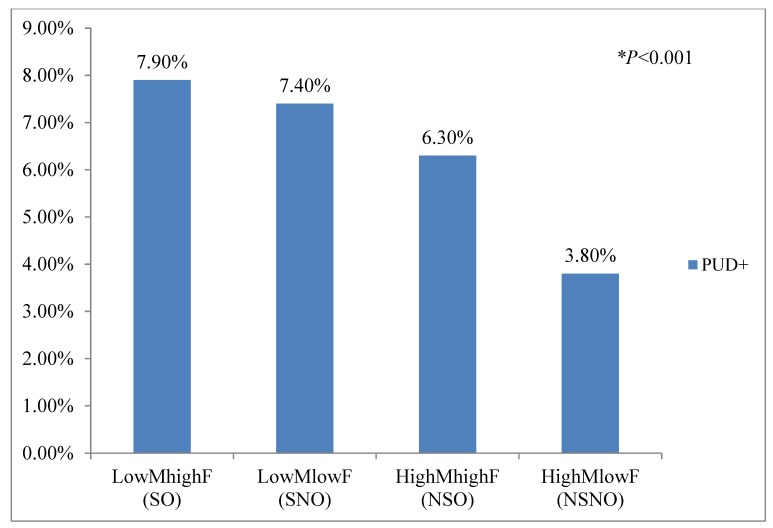
Prevalence of PUD according to muscle mass and fat mass. Abbreviations: LowMhighF, low-muscle high-fat group; HighMlowF, high-muscle low-fat group; HighMhighF, high-muscle high-fat group; HighMlowF, high-muscle low-fat group; SO group, sarcopenic obesity; SNO group, sarcopenic non-obesity; NSO group, non-sarcopenic obesity; NSNO, non-sarcopenic non-obesity; PUD, peptic ulcer disease. * Linear by linear *p* < 0.001.

**Table 1 medicina-56-00121-t001:** General characteristics of the patients according to the presence of sarcopenia.

	Sarcopenia-(n = 3915)	Sarcopenia + (n = 3177)	*p*
Age, mean ± SD	48.6 ± 16.3	49.8 ± 16.9	0.001
Male, n (%)	1698 (43.4%)	1376 (43.3%)	0.9
Education (< elementary school)	582 (14.9%)	1378 (43.4%)	<0.001
Household income (quantile) (1st (lowest))	534 (13.1%)	963 (30.3%)	<0.001
Existence of spouse, N (%)	3077 (78.6%)	2971 (93.5%)	<0.001
**Health Behaviors**			
Current drinker, N (%)	2765 (70.6%)	1673 (52.7%)	<0.001
Current smoker, N (%)	2283 (58.3%)	1860 (58.5%)	0.2
Regular exercise, N (%)	1801 (46.0%)	1408 (44.3%)	0.2
**Medical History**			
Hypertension, N (%)	595 (15.2%)	497 (15.6%)	0.6
Diabetes mellitus, N (%)	3626 (92.6%)	2817 (88.7%)	<0.001
Cardiovascular disease with medication, N (%)	74 (1.9%)	198 (6.2%)	<0.001
Musculoskeletal pain with medication, N (%)	389 (9.9%)	811 (25.5%)	<0.001
Depression, N (%)	557 (14.2%)	585 (18.4%)	<0.001
Stress, N (%)	1213 (31.1%)	893 (28.1%)	0.009
**Laboratory Data**			
Fasting blood glucose (mg/dL)	94.1 ± 19.5	102.5 ± 27.8	<0.001
Triglyceride, mg/dL	119.8 ± 100.3	153.7 ± 116.9	<0.001
LDL, mg/dL	107.6 ± 28.8	116.5 ± 32.4	<0.001
**BMI, kg/m^2^**	22.7 ± 3.1	24.8 ± 3.3	<0.001
**Waist circumference (cm)**	78.4 ± 9.5	84.3 ± 9.5	<0.001
**Peptic ulcer disease**	216 (5.5%)	240 (7.6%)	0.001

Data are means ± Standard deviation (SD) or percentages; Abbreviations: HbA1c, hemoglobin A1c; LDL, low-density lipoprotein; HDL, high-density lipoprotein; BMI, body mass index.

**Table 2 medicina-56-00121-t002:** General characteristics according to the presence of peptic ulcer disease.

	PUD-(N = 6636)	PUD + (N = 456)	*p*
Age, mean age (SD)	48.7 ± 16.7	56.6 ± 14.1	<0.001
Male n (%)	2841 (42.8%)	233 (51.1%)	<0.001
Education (< Elementary school)	1782 (26.9%)	178 (39.0%)	<0.001
Household income (quantile) (1st (lowest))	1645 (24.8%)	107 (23.5%)	0.3
Existence of spouse	5620 (84.7%)	438 (96.1%)	<0.001
**Health Behaviors**			
Current drinker, N (%)	4165 (62.8%)	273 (59.9%)	0.4
Current smoker, N (%)	3912 (59.0%)	231 (50.7%)	<0.001
Regular exercise, N (%)	3002 (45.2%)	207 (45.4%)	1.0
**Medical History**			
Hypertension, N (%)	5607 (84.5%)	393 (86.2%)	0.3
Hyperlipidemia, N (%)	3424 (51.6%)	241 (52.9%)	0.6
Diabetes mellitus, N (%)	6036 (91.0%)	407 (89.3%)	0.2
Cardiovascular disease with medication, N (%)	235 (3.5%)	37 (8.1%)	<0.001
Musculoskeletal pain with medication, N (%)	1087 (16.4%)	113 (24.8%)	<0.001
Depression, N (%)	1038 (15.6%)	104 (22.8%)	<0.001
Stress, N (%)	1939 (29.2%)	167 (36.6%)	0.001
**Laboratory Data**			
Fasting blood glucose	97.7 ± 24.1	98.5 ± 19.9	0.1
HbA1C,%	7.3 ± 1.4	7.0 ± 1.1	0.06
Insulin, uIU/mL	9.8 ± 6.2	9.3 ± 4.3	0.2
Total cholesterol, mg/dL	186.5 ± 35.4	189.0 ± 34.3	1.6
Triglyceride, mg/dL	133.9 ± 107.6	147.3 ± 129.6	0.01
LDL, mg/dL	111.1 ± 30.7	112.9 ± 30.2	0.6

Data are means ± SE or percentages (SE); Abbreviations: HbA1c, hemoglobin A1c; LDL, low-density lipoprotein.

**Table 3 medicina-56-00121-t003:** Obesity and sarcopenic indices according to the presence of peptic ulcer disease.

	PUD-(N = 6636)	PUD + (N = 456)	*p*
**Body mass index (kg/m^2^)**	23.7 ± 3.4	23.6 ± 3.2	0.8
**Waist circumference (cm)**	81.0 ± 10.0	82.2 ± 9.0	0.008
**Appendiceal muscle mass index BMI**			0.001
Normal to high	3699 (55.7%)	216 (47.4%)	
Low	2937 (44.3%)	240 (52.6%)	
**Appendiceal muscle mass index weight (kg)**	30.2 ± 4.4	30.6 ± 4.6	0.06
Normal to high	4655 (70.1%)	301 (66.0%)	
Low	1981 (29.9%)	155 (34.0%)	
**Appendiceal muscle mass index height 2(m^2^)**	7.1 ± 1.3	7.1 ± 1.2	0.4
Normal to high	5738 (86.5%)	379 (83.1%)	
Low	898 (13.5%)	77 (16.9%)	
**Fat mass (kg/total body)**	3.4 ± 1.4	3.3 ± 1.4	0.8
**Fat muscle mass index**			<0.001
LowMhighF (sarcopenic obesity)	813 (12.3%)	70 (15.4%)	
LowMlowF (sarcopenic non-obesity)	2124 (32.0%)	170 (37.3%)	
HighMhighF (non-sarcopenic obesity)	2494 (37.6%)	169 (37.1%)	
HighMlowF (non-sarcopenic non-obesity)	1205 (18.2%)	47 (10.3%)	

Data are means ± SE or percentages (SE); Abbreviations: LowMhighF, low-muscle high-fat group; HighMlowF, high-muscle low-fat group; HighMhighF, high-muscle high-fat group; HighMlowF, high-muscle low-fat group.

**Table 4 medicina-56-00121-t004:** Odds ratios for risk factors for peptic ulcer disease.

		SO Group (LowMhighF) (N = 883)	SNO Group (LowMlowF) (N = 2294)	NSO Group (HighMhighF) (N = 2663)	NSNO Group (HighMlowF) (N = 1252)
PUD+	Proportion n (%)	70 (7.9%)	170 (7.4%)	169 (6.3%)	47 (3.8%)
	Crude OR	2.2 (1.5–3.2) ***	2.1 (1.5–2.9) ***	1.7 (1.2–2.4) ***	1 (reference)
	Age, sex OR	1.9 (1.3–2.6) ***	1.4 (1.0–2.0) **	1.3 (0.9–2.0)	1 (reference)
	**Model 1**	1.9 (1.3–2.7) ***	1.5 (1.0–2.0) **	1.4 (0.9–2.1)	1 (reference)
	**Model 2**	1.9 (1.3–2.7) ***	1.4 (1.0–2.0) *	1.3 (0.9–2.1)	1 (reference)
	**Model 3**	1.7 (1.2–2.5) **	1.4 (0.9–1.9)	1.2 (0.8–1.9)	1 (reference)
	**Model 4**	1.7 (1.1–2.4) **	1.3 (0.9–1.9)	1.2 (0.7–1.8)	1 (reference)
	**Model 5**	1.7 (1.2–2.5) **	1.3 (0.9–1.9)	1.2 (0.7–1.8)	1 (reference)
	**Model 6**	1.7 (1.2–2.5) **	1.3 (0.9–1.9)	1.1 (0.7–1.8)	1 (reference)

Model 1: Adjusted for age, sex, and BMI; Model 2: Adjusted for age, sex, BMI, and HOMA-IR (homeostatic model assessment insulin resistance); Model 3: Adjusted for age, sex, BMI, HOMA-IR, smoking, and drinking; Model 4: Adjusted for age, sex, BMI, HOMA-IR, smoking, drinking, medication for cardiovascular disorder, and medication for muscular skeletal disorder; Model 5: Adjusted for age, sex, BMI, HOMA-IR, smoking, drinking, medication for cardiovascular disorder, medication for muscular skeletal disorder, exercise, education, income, and spouse; Model 6: Adjusted for age, sex, BMI, HOMA-IR, smoking, drinking, medication for cardiovascular disorder, medication for muscular skeletal disorder, exercise, education, income, spouse, stress, and depressive symptoms. Each muscle mass and fat mass category was compared to the HighMlowF group. * *p* < 0.05, ** *p* < 0.01, *** *p* < 0.001 compared to the HighMlowF group. Abbreviations: LowMhighF, low-muscle high-fat group; HighMlowF, high-muscle low-fat group; HighMhighF, high-muscle high-fat group; HighMlowF, high-muscle low-fat group; SO group, sarcopenic obesity; SNO group, sarcopenic non-obesity; NSO group, non-sarcopenic obesity; NSNO, non-sarcopenic non-obesity.

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
