# Peer review of "Sarcopenia is Independently Associated with an Increased Risk of Peptic Ulcer Disease: A Nationwide Population-Based Study"

_medicina, 2020, doi:10.3390/medicina56030121_

Round 1
Reviewer 1 Report
Dear Authors,
in this review, you aim to present the association of sarcopenia and obesity with peptic ulcer disease.
The topic is interesting and the article is well written.
On the other hand, I suggest some minor revisions to improve this work.
Minor revisions
Page 2 Lines 44-47 “Sarcopenia has been related with diverse disease outcomes including cardiovascular disease, liver disease, lung and kidney disease, frailty, cognitive dysfunction, depression, and almost all types of malignancies”: Please improve this part taking into account how Inflammatory Bowel Diseases might also be a cause of secondary sarcopenia, citing the article “Pizzoferrato, M.; de Sire, R.; Ingravalle, F.; Mentella, M.C.; Petito, V.; Martone, A.M.; Landi, F.; Miggiano, G.A.D.; Mele, M.C.; Lopetuso, L.R.; Schiavoni, E.; Napolitano, D.; Turchini, L.; Poscia, A.; Nicolotti, N.; Papa, A.; Armuzzi, A.; Scaldaferri, F.; Gasbarrini, A. Characterization of Sarcopenia in an IBD Population Attending an Italian Gastroenterology Tertiary Center. Nutrients 2019, 11, 2281”.
Reviewer 2 Report
As the author states, a few studies have shown that sarcopenia is a factor related to PUD, and I think this study is useful. However, the following points are not clear, so please consider making the following revisions.
Methods
This manuscript states that the definition of sarcopenia was based on the one recommended by the AWGS, but I think this explanation is wrong. Motor function parameters such as hand-grip strength and gait speed should be examined first, rather than muscle mass. Further, the criteria for sarcopenia should be reexamined because the criterion of skeletal muscle mass is not applicable to young adults.
In the criterion for skeletal muscle mass, I do not understand why the author used “Appendiceal muscle mass index_BMI” and “Appendiceal muscle mass index_weight” as ASM. Please provide the reason for this.
Results
In the multiple regression analyses, the author compared four groups. However, in the results section, the author compared two groups. I think these comparisons are incongruous with each other. Should you not compare four groups and clarify the characteristics of each group?
In Table 1, the author mentioned “Age, mean age (SD) ≧55 years” and the number of participants, but I do not understand the meaning. Conventionally, the average age should be mentioned as it is. In addition, why was the age set to 55 years or older? The author should provide reasons for this (the AWGS recommends using the age of 60 years or older for sarcopenia diagnosis).
If you are evaluating the participants for sarcopenia, body composition (ASM, fat mass, and so on) should be described in detail.
In Table 3, body composition differs between men and women. Therefore, the results for each gender should be listed.
Round 2
Reviewer 2 Report
Your manuscript has been almost corrected.
I think it is no problem.
Author Response
Reveiwer 2's comments: Your manuscript has been almost corrected.
I think it is no problem.
(Author reply)
Dear Reviewers,
I would like to thank the editor and reviewers of Medicina for their review of our article. The reviewers’ comments enabled us to revise our manuscript and improve the quality of it.
